Genetic variation and selection in the major histocompatibility complex Class II gene in the Guizhou pony

Liu Chang 1 2
Lei Hongmei 1
Ran Xueqin xqran@gzu.edu.cn 1
Wang Jiafu jfwang@gzu.edu.cn 1 3
1 College of Animal Sciences, Guizhou University , Guiyang , China
2 College of Pharmacy, Guizhou University of Traditional Chinese Medicine , Guiyang , China
3 Tongren University , Tongren , China
Harrison Xavier
Electronic publication date: 2020 Sep 18
Publication date: 2020
Volume: 8
Electronic Location ID: e9889
Received 2019 Nov 25; Accepted 2020 Aug 17
Copyright: ©2020 Liu et al.
Copyright year: 2020
Copyright holder: Liu et al.
License: This is an open access article distributed under the terms of the Creative Commons Attribution License, which permits unrestricted use, distribution, reproduction and adaptation in any medium and for any purpose provided that it is properly attributed. For attribution, the original author(s), title, publication source (PeerJ) and either DOI or URL of the article must be cited.
License URL: https://creativecommons.org/licenses/by/4.0/

Keywords: Guizhou pony, MHC, Antigen binding sites, Evolution process, Adaptation

Funding: The Talents of Guizhou Science and Technology Cooperation Platform QKHPTRC[2019]-5615 The Guizhou Province Hundred Innovative Talents Project QKHRC[2016]-4012] The Guizhou Agriculture Research program QKHNYZ[2012]3009 QKHNYZ[2009]3068 This work was supported by The Talents of Guizhou Science and Technology Cooperation Platform (QKHPTRC[2019]-5615), The Guizhou Province Hundred Innovative Talents Project (QKHRC[2016]-4012], and the Guizhou Agriculture Research program (QKHNYZ[2012]3009, QKHNYZ[2009]3068). The funders had no role in study design, data collection and analysis, decision to publish, or preparation of the manuscript.

==============================
The Guizhou pony (GZP) is an indigenous species of equid found in the mountains of the Guizhou province in southwest China. We selected four regions of the equine leukocyte antigen (ELA), including DQA, DRA, DQB, and DRB, and used them to assess the diversity of the major histocompatibility complex (MHC) class II gene using direct sequencing technology. DRA had the lowest dN/dS ratio (0.560) compared with the other three loci, indicating that DRA was conserved and could be conserved after undergoing selective processes. Nine DQA, five DQB, nine DRA, and seven DRB codons were under significant positive selection at the antigen binding sites (ABS), suggesting that the selected residues in ABS may play a significant role in the innate immune system of the GZP. Two GZP alleles were shared with Przewalski’s horse, and six older GZP haplotypes had a better relationship with other horse species by one or two mutational steps, indicating that the GZP may be a natural ancient variety of equid. The specific diversity of ABS and the numbers of unique haplotypes in the evolutionary process affords this species a better genetic fitness and ability to adapt to the native environment.

Introduction

The major histocompatibility complex (MHC) genes play a major role in vertebrate immune systems and have a high degree of genetic diversity associated with the adaptive immune response and evolution (Lian et al., 2017; Kamath & Getz, 2011). The MHC system is divided into class I and class II, which are key parts of the immune system (Hughes & Nei, 1988). The MHC class II genes are highly polymorphic parts of the immune response that act by presenting extracellular antigens to T lymphocytes. These molecules are heterodimers with α and β chains encoded by A and B genes. The polymorphic sites of the class II genes are typically located at exon 2, which codes for the first extracellular domain or the antigen binding sites (ABS). The exon 2 codes for a section of the pocket of the MHC molecule. The ABS mainly encoded the second exon of the MHC class II gene and have more variation than the neighboring regions in this sequence (Li et al., 2014), indicating that ABS variation may help to determine the rates of evolution across the MHC (Hughes & Hughes, 1995). Previous studies have shown that exon 2 of MHC class II genes had the most polymorphisms and encoded the α and β domains principally responsible for peptide binding (O’Connor et al., 2007). The polymorphism of the MHC loci is commonly associated with different susceptibilities to infectious diseases (Hill, 2001), especially in sheep (Paterson, Wilson & Pemberton, 1998), mice (Meyer-Lucht & Sommer, 2005), voles (Kloch et al., 2010) and lemurs (Schad, Ganzhorn & Sommer, 2005). The equine MHC class II loci may assist in determining the host response to pathogens encountered by the horse (Miller et al., 2017). MHC variants play key roles in mate preference, kin recognition, and maternal-fetal interactions (Edwards & Hedrick, 1988; Bernatchez & Landry, 2003; Piertney & Oliver, 2006). The diverse functions and characteristics of MHC molecules is reflected in the evolutionary and adaptive processes within and between populations (Sommer, 2005).

The mechanisms of negative frequency-dependent selection (NFDS) and over-dominant selection have been well-studied in MHC genes. NFDS maintains intraspecific diversity and may interact with population density (Levitan & Ferrell, 2006; Meyer & Kassen, 2007). Over-dominant selection can maintain genetic polymorphisms in populations (Takahata & Nei, 1990). Correlative and experimental support for the negative frequency-dependent selection of MHC genes has been shown in humans (Trachtenberg et al., 2003), reed warblers (Westerdahl et al., 2004), mice (Kubinak et al., 2012), sticklebacks (Eizaguirre et al., 2012; Bolnick & Stutz, 2017) and guppies (Phillips et al., 2018). There are a number of examples of asymmetric over-dominant selection in populations found in the wild and in the laboratory (Landry & Bernatchez, 2001; Richman, Herrera & Nash, 2001; Lenz et al., 2009; Schwensow et al., 2010; Lenz et al., 2013). These results are supported by several computer-based binding prediction studies (Lenz, 2011; Lau et al., 2015; Buhler, Nunes & Sanchez-Mazas, 2016; Pierini & Lenz, 2018). Three primary sources of evidence currently support the idea of balancing selection: (i) elevated levels of polymorphisms, (ii) the rates of nonsynonymous (dN) to synonymous (dS) nucleotide substitutions (Hughes & Nei, 1988; Hughes & Nei, 1989), and (iii) trans-species polymorphisms with alleles among species (Klein et al., 1993). The dN/dS ratio is frequently used to measure selective pressure on genes (Yang et al., 2000), and more specifically, the markedly different rates of evolution across the MHC genes (Hughes & Hughes, 1995). Site-specific methods have found elevated dN/dS ratios at ABS, suggesting substantially different rates of evolution across the MHC. MHC variation within species and among species has proven to be useful in determining the historical patterns of selection in various mammals (Cutrera & Lacey, 2007).

In the family Equidae, the horse MHC class II gene, also known as equine leukocyte antigen (ELA) class II, is located on the short arm of chromosome 20q14-q22 (Mäkinen et al., 1989; Ansari et al., 1988). It contains the DQA, DQB, DRA, and DRB genes. The DQA and DRA genes encode for the α-chain of ELA class II molecules, and the polymorphisms of the DQA and DRA genes have been determined in European equids (Luís et al., 2005; Janova et al., 2009; Kamath & Getz, 2011). The DQB and DRB genes encode the β-chain of the ELA class II complex, and high levels of DRB and DQB polymorphisms have been reported in Arabian and European horses (Fraser & Bailey, 1996; Mashima, 2003). Previous reports indicated that exon 2 of the ELA class II gene is genetically diverse among horse populations (Kamath & Getz, 2011). We examined the sequence variation in the second exon to determine the selective pressures and evolutionary path for the Guizhou pony.

The Guizhou pony is an indigenous species that was found in the Guizhou province during the Warring States Period (475-221 B.C.) in Ancient China. It is one of five Chinese pony species and has a body height of only 1.1 m (10-11 hands). The mtDNA/SSR polymorphism has been determined in several pony populations derived from native Irish, Canadian, and Chinese breeds using mtDNA/SSR markers (McGahern et al., 2006; Prystupa et al., 2012). We sought to analyze the variation in the MHC II exon 2 of the DQA, DRA, DQB, and DRB regions and their relationship with the selection and evolution in the GZP.

Material and Methods

Animal collection and DNA isolation

A total of 50 blood samples were collected from GZP in Ziyun County, Anshun, Guizhou Province, China. All ponies used in our study were 4 to 8 years old. All animal procedures were approved by the Institutional Animal Care and Use Committee of Guizhou University (Approval number EAE-GZU-2018-P007). The GZP were randomly selected and were all well-developed and in good health, with heights ranging from 102 to 118 cm and weights between 210 to 265 kg. Blood samples were collected from the jugular vein and were kept in EDTA Na2. All samples were stored at −20 °C until DNA extraction. Genomic DNA was extracted from blood samples using the SQ Blood DNA Kit (OMEGA, USA). The nucleic acid concentration of the extracted genomic DNA was calculated by determining OD260/OD280, and detected by 0.7% agarose gel electrophoresis.

Table 1 The primers for DRA, DRB, DQA and DQB gene detection.

Gene name	Primer name	Primer sequence (5′→3′)	Length (bp)	
DRAexon2	DRA-F	AGGATCACGTGATCATCCAG	246	
DRA-R	CATTGGTGTTTGGAGTGTTG	
DRBexon2	DRB-F	CTCTGCAGCACATTTCCTGGAG	276	
DRB-R	CGCCGCTGCACCAGGAA	
DQBexon2	DQB-F	CTCGGATCCGCATGTGCTACTTCACCAACG	230	
DQB-R	GAGCTGACGGTAGTTGTGTCTGCACAC	
DQAexon2	DQA-F	CTGATCACTTTGCCTCCTATG	246	
DQA-R	TGGTAGCAGCAGTAGAGTTG	

PCR amplification, cloning, and sequencing

The exon 2 regions of the ELA-DQA, DQB, DRA, and DRB genes were amplified from genomic DNA using PCR with specific primers. We amplified 246 bp of the DRA using the equid-specific primers DRA-F and DRA-R (Albright-Fraser et al., 1996), 246 bp of the DQA using the primers DQA-F and DQA-R (Fraser & Bailey, 1998), 276 bp of the DRB using the primers DRB-F and DRB-R (Fraser & Bailey, 1996), and 230 bp of the DQB using the primers DQB-F and DQB-R (Mashima, 2003). All primers were synthesized by the Bio-Engineering Company (Shanghai, China) (Table 1). The total PCR volume was 20 µL, and contained 10 µL of 2 × PCR Mixture (0.1 U Taq Plus Polymerase/µL, 500 µM dNTP each, 20 mM Tris–HCl (pH8.3), 100 mM KCl, 3 mM MgCl2), 0.4 µL of upstream/downstream primers (10 µmol/L), and 1 µL templates. PCR amplification was carried out with initial denaturation at 95 °C for 5 min, followed by 30 cycles (95 °C for 30 s, 58 °C for 30 s, and 72 °C for 30 s), and a final extension at 72 °C for 10 min. PCR products were extracted and purified using the Gel Extraction Kit (OMEGA, USA), and were ligated into pGEM®-T vectors and transformed into E. coli competent cells. Twenty positive clones of each sample were removed with a sterile toothpick and were detected using the Sanger sequencing method (Invitrogen, China). Alleles were confirmed if the same allele was observed in at least two different individuals.

DNA sequence polymorphism analysis

The base composition of the DRA, DRB, DQA and DQB genes was analyzed using MEGA7 software (Kumar, Stecher & Tamura, 2016). Standard descriptive diversity indices for each locus within the GZP were calculated using MEGA7 software, including the variable sites (V), parsim-info sites (P), singleton sites (S), and the transition/transversion bias ratio (R). It was important to ascertain whether the variability was uniformly distributed or was confined to small segments of the variable regions when determining the nature of the variable region. The Wu–Kabat variability index was calculated using the formula by Wu and Kabat Wu & Kabat (1970) with respect to amino acids at peptide-binding pockets. The variation of amino acids was calculated by the mutation rate (variability = number of different amino acids at a certain position/frequency of the most common amino acids at this position) (Wu & Kabat, 1970). Selection was estimated using MEGA7 software in terms of the relative rates of nonsynonymous (dN) and synonymous (dS) mutations, according to Nei and Gojobori’s method with the Jukes and Cantor (JC) correction (Nei & Gojobori, 1986). The selection Z-Test (P < 0.05) was performed for all sites under the null hypothesis of neutrality (dN = dS) and the alternative hypotheses of non-neutrality (dN ≠ dS), positive selection (dN >  dS), and purifying selection (dN <  dS).

Site-specific selection analyses and protein 3D structure analysis

We estimated the nonsynonymous and synonymous substitutions in the overall domain, ABS, and non-ABS for the DQA, DQB, DRA and DRB alleles. We assessed the positive selection using CodeML subroutine in the PAML program (Yang, 2007), which was more sensitive than other methods for assessing selection at the molecular level (Anisimova, Bielawski & Yang, 2001). The PAML program used the maximum likelihood estimation to examine heterogeneity in rates of ω = dN/dS among codons (Bielawski & Yang, 2003). The PAML program was able to better detect the molecular evidence of selection compared to other programs (Anisimova, Nielsen & Yang, 2003). We assessed heterogeneity in ω (ω  <  1: purifying selection, ω = 1: neutral evolution, ω >  1: positive selection) across the four alleles (DQA, DQB, DRA and DRB) to identify codons under positive selection. The observed ω value followed six models in PAML: M0 (one ratio, average ω across all sites), M1a (nearly neutral), M2a (positive selection), M3 (discrete), M7 (beta), and M8 (beta and omega) (Yang et al., 2000). We used the online SWISS-MODEL program (Biasini et al., 2014; Waterhouse et al., 2018) (https://swissmodel.expasy.org/interactive) to make predictions about the DQA, DQB, DRA and DRB protein structures.

Phylogenetic allele networks

We constructed a median-joining haplotype network to infer the phylogenetic relationships among the sequence haplotypes (Bandelt, Forster & Röhl, 1999) using the maximum parsimony in Network 4.6.1 (http://www.fluxus-engineering.com/sharenet.htm). The haplotype median networks of DQA, DQB, DRA and DRB between GZP and known horse species (Eqca, E.callabus; Eqpr, E.przewalski; Eqki, E.kiang; Eqgr, E.grevyi; Eqas, E.asinus; Eqbu, E.burchelli; Eqze, E.zebra; Eqhe, E.hemionus) from GenBank were plotted using NetWork 4.6. The frequency information and population proportion of the alleles were incorporated into the visualization of the network. Sequences from the horse, including E. callabus, E. przewalskii, E. burchellii, E. asinus, were incorporated to evaluate the distance from the Guizhou pony’s haplotypes (Table 2).

Results

Analysis of nucleotide diversity

184 alleles were identified from 1,000 sequencing clones, with 118 effective alleles selected from the total. Of the 118 alleles, there were 18 novel DQA alleles (GenBank accession number: MT304744 –MT304761), 38 new DQB alleles (MT304705 –MT304743), 22 new DRA alleles (MT304762 –MT304783) and 28 new DRB alleles (MT304784 –MT304811) (Table S1). The alignment results are listed in Table S1 for the effective alleles from DQA, DQB, DRA, DRB and the sequences of JQ254059, AF034122, AJ575295, and AF144564. A considerable sequence diversity within the genus was revealed based on the DQA, DQB, DRB alignment results. The nucleotide diversity in DRA was much lower than in DQA/B and DRB in GZP, which is comparable with the level of nucleotide diversity in DRA from other species in the Equus genus (Kamath & Getz, 2011). Within the GZP, the genetic diversity was much higher in DQA, DQB, and DRB than in DRA and the ratio (variable sites/length) was the lowest at the DRA locus (15.04%) and highest at the DQB locus (46.08%).

Analysis of nucleotide compositions

The GC contents of DQB and DRB were higher than those of DQA and DRA (Table S2). The content of G+C (48.10%) was slightly lower than that of A+T (51.90%) at DQA, and the content of G+C (48.10%) was lower than that of A+T (51.90%) at DRA, which revealed that DQA and DRA had lower GC percentages. However, the base composition of G+C (64.20%) was higher than that of A+T (35.80%) in DQB alleles, and the base composition of G+C (61.90%) was more than that of A+T (38.10%) in DRB alleles, revealing that DQB and DRB had a higher GC content. The R (transitions/transversions) was 1.357 and 2.241 in the DQA and DRA alleles, respectively. However, there was an R of 0.778 and 0.573 in the DQB and DRB alleles, respectively. Our results revealed that the DRA locus was more well-conserved than the other loci.

Table 2 General information regarding the horse populations analyzed in this Network study.

Locus	Breed	Source	GenBank ID	
DQA	Guizhou pony	This study	MT304744 –MT304761	
DQA	Equus przewalskii	NCBI	JX088699, JX088698, JX088697, JX088696, U92509, U92509	
DQA	Equus caballus	NCBI	AF115329, AF115328, AF115327, AF115326, AF115325, AF115324, U92508, U92519,
U92518, U92517, U92516, U92515,
U92514, U92513, U92512, U92511,
U92510, U92507, U92506, U92505	
DQA	Equus burchellii	NCBI	EU935835, EU935837, EU935836, EU935834,
EU935833, EU935832, EU935829, EU930130
HQ637409, HQ637408, HQ637407,
HQ637397, HQ637406, HQ637405,
HQ637404, HQ637403, HQ637402,
HQ637401, HQ637400, HQ637399,
HQ637398	
DQA	Equus zebra	NCBI	EU935838, EU935831, EU935830, EU935828	
DQA	Equus grevyi	NCBI	EU930136, EU930131	
DQA	Equus asinus	NCBI	U92522, U92521	
DQA	Equus hemionus	NCBI	U92520, EU930135	
DQA	Equus kiang	NCBI	EU930134, EU930133, EU930132	
DQB	Guizhou pony	This study	MT304705 –MT304743	
DQB	Equus asinus	NCBI	AF034125, AF034124, AF034123, AF034122,
U31776, U31775, U31774, XM_014839831.1	
DQB	Equus przewalskii	NCBI	XM_008508365.1	
DQB	Equus caballus	NCBI	JQ254069.1, L33910.1, XM_005603501.3, JQ254075.1, JQ254071.1, NM_001317256.1	
DRA	Guizhou pony	This study	MT304762 –MT304783	
DRA	Equus hemionus	NCBI	L47173, EU930128	
DRA	Equus kiang	NCBI	FJ657514, EU930127	
DRA	Equus grevyi	NCBI	EU930125, EU930116	
DRA	Equus zebra	NCBI	EU930117, EU930119, EU930123, EU930124,
EU930129	
DRA	Equus burchellii	NCBI	EU930118, EU930120, EU930121, EU930122, EU930126, HQ637392, HQ637393,
HQ637394, HQ637396, HQ637395,
AJ575299	
DRA	Equus caballus	NCBI	L47172, L47174, AJ575295, JN035631,
JN035630, JN035629	
DRA	Equus asinus		AJ575298, AJ575297, AJ575296, HM165492,
FJ487912, L47171, AF541938	
DRB	Guizhou pony	This study	MT304784 –MT304811	
DRB	Equus przewalskii	NCBI	AF084188.1, XM_008511984.1	
DRB	Equus caballus	NCBI	L76972, L76978, L76976, L76975, L76974,
L76977, L76973, AF170067, L77079, AF144564, XM_023624024.1, JQ254087.1
XR_002801945.1, XR_001379170.2,
XM_023624023.1, XM_014734203.2,
XM_014734205.1, NM_001142816.1,
JN035625.1, JQ254096.1, JQ254095.1,
L25644.1, JN035627, JN035623.1,
JN035622.1, JN035621.1, JN035624.1,
JQ254099.1, JN035626.1, JQ254093.1	
DRB	Equus asinus	NCBI	XM_014846373.1, XR_001398881.1,
XM_014846372.1, KJ596517.1, KJ596507.1,
KJ596519.1, KJ596510.1, KJ596510.1,
KJ596516.1, KJ596511.1, KJ596518.1,
KJ596512.1, KJ596514.1, KJ596515.1	

The amino acid composition analyses

We determined that the exon 2 of the DQA, DQB, DRA, and DRB nucleotide sequences encoded 82, 76, 81, and 79 amino acid sequences, respectively. The underlined residues in Fig. 1 indicated an assumed ABS, based on the HLA equivalents (Brown et al., 1988; Brown et al., 1993), and may contact the antigen peptides (Figs. 1A–1D). There were 38 (45.12%), 51 (67.10%), 16 (19.75%), and 49 sites (62.02%) that were variable in the predicted amino acid sites of the DQA, DQB, DRA and DRB of GZP populations, respectively. For the ABS, 17 of 21 sites (80.95%), 12 of 18 sites (66.67%), 5 of 20 sites (25.00%), and 13 of 14 sites (92.85%) were diverse at the DQA, DQA, DQB, DRA and DRB loci. The amino acid compositions of DQA, DQB, DRA, and DRB at ABS were calculated using MEGA 7 software (Fig. 2). There were more polar R-amino acids at the DQA locus (46.30%), and included Gly, Cys, Ser, Tyr, Thr, Asn, and Gln (Fig. 2). The non-polar R-amino acids at the DRA locus had the highest percentage (58.86%), and included Ala, Leu, Val, Trp, Ile, Phe, Pro, and Met (Fig. 2). The largest proportion of charged R-amino acids (30.70%) was located at DQB, and consisted of Arg, Lys, His, Glu, and Asp (Fig. 2).

Figure 1 The amino acid alignment of the DQA (A), DQB (B), DRA (C) and DRB (D) locus.

Underlines below amino acids indicated antigen binding sites (ABS). The missing amino acid was denoted with hyphen.

Figure 2 The distribution of amino acids with non-polar, polar and positively, negatively charged residues.

Non-polar R-amino acids: Ala, Leu, Val, Trp, Ile, Phe, Pro, Met; polar R-amino acids: Gly, Cys, Ser, Tyr, Thr, Asn, Gln; charged R-amino acids: Arg, Lys, His, Glu, Asp.

Global selection analyses

The Wu-Kabat variability index was not used to select all of the variable amino acids (Wu & Kabat, 1970). A total of fifteen amino acids at the DQA locus were strongly selected at residues 10, 17, 18, 21, 23, 30, 46, 51, 52, 58, 60, 61, 62, 65 and 72, with the highest variability occurring at residue 60 (Fig. 3). Many polymorphic sites were observed at the DQB locus, with eight high mutation loci at residues at 16, 27, 38, 46, 47, 57, 61, and 65 (Fig. 3). 12 residues were found to be polymorphic at the DRA locus, including residues at 12, 14, 15, 19, 29, 39, 47, 49, 63, 64, 67 and 69 (Fig. 3). Amino acid residues at six different positions in the DRB locus had high values (more than 30) on the Wu-Kabat variability index, and the strongly selected amino acid regions were found at residues 1, 2, 4, 5, 6, 7, 8, 12, 19, 28, 36, 47, 48, 50, 56, 58, 61, 62, 65 and 69 (Fig. 3). A comparison of the dN/dS ratio averaged across the whole coding region suggested that positive selection occurred at loci DQB (dN/ dS = 1.127, p = 0.322) and DRB (dN/dS = 1.228, p = 0.202), and purifying selection appeared at DQA (dN/dS = 0.779, p = 0.143) and DRA (dN/dS = 0.560, p = 0.069) (Table 3). All codon sites were not statistically significant according to the Z-tests (p > 0.05, Table 3). The estimates of dN/dS suggested that DQA and DRA were not affected by the positive selection at the genetic level.

Figure 3 Wu-Kabat variability index of DQA (A), DQB (B), DRA (C) and DRB (D) loci..

The vertical index indicated the Wu-Kabat index at each amino acid position. The horizontal axis showed the consensus amino acids in the DQA, DQB, DRA and DRB peptides.

Site-specific selection analyses

It is unlikely for selection to act uniformly across MHC genes over evolutionary time. Selection was more likely to occur at specific codons based on their functional role. The rate of nonsynonymous substitutions for the ABS (dN = 0.594 ± 0.132) exceeded the number of synonymous substitutions four times (dS = 0.128 ±  0.080) at the DRB (Table 3). Our results are in agreement with those observed in the Argentine Creole horse, which exhibited rates of nonsynonymous substitutions more than four times the number of synonymous substitutions at exon 2 of ELA-DRB (Díaz et al., 2001). The ABS rates of synonymous substitutions and nonsynonymous substitutions for DQA and DQB were similar (dN = 0.330 ± 0.088, dS = 0.287 ± 0.110; dN = 0.206 ± 0.079, dS = 0.133 ±  0.076, respectively) (Table 3). The ABS sites at the DRA exhibited less nonsynonymous substitutions (dN = 0.017 ±  0.008) than synonymous substitutions (dS = 0.028 ± 0.022) with a dN/dS ratio of 0.607 (Table 3). Z-tests performed separately on ABS were significant for DRB (p = 0.001) providing evidence for positive selection at these sites. We could not reject the null hypothesis of neutral evolution at the non-ABS site (Table 3). The Z-tests by site type at the DQA, DQB, and DRA sites could not reject the null hypothesis of neutrality (p > 0.05). In contrast, the non-ABS sites showed more synonymous substitutions than nonsynonymous substitutions with dN/dS ratios of 0.581, 0.895, 0.520, and 0.678 at DQA, DQB, DRA and DRB, respectively (Table 3).

Table 3 The Indices of selection at the DQA, DQB, DRA and DRB loci.

Allele	Type	No.	aa distance	dN	dS	dN/ dS	Z;
dN≠dS	Z;
dN > dS	Z;
dN < dS	
DQA	All	82	0.213 ± 0.029	0.138 ± 0.024	0.177 ± 0.042	0.779	0.296	1.000	0.143	
	ABS	21	0.416 ± 0.066	0.330 ± 0.088	0.287 ± 0.110	1.150	0.728	0.361	1.000	
	non-ABS	61	0.148 ± 0.031	0.086 ± 0.021	0.148 ± 0.041	0.581	0.123	1.000	0.061	
DQB	All	76	0.155 ± 0.026	0.097 ± 0.018	0.086 ± 0.018	1.127	0.621	0.322	1.000	
	ABS	19	0.264 ± 0.071	0.206 ± 0.079	0.133 ± 0.076	1.549	0.248	0.123	1.000	
	non-ABS	57	0.124 ± 0.024	0.068 ± 0.014	0.076 ± 0019	0.895	0.734	1.000	0.371	
DRA	All	81	0.028 ± 0.007	0.014 ± 0.004	0.025 ± 0.007	0.560	0.138	1.000	0.069	
	ABS	20	0.042 ± 0.019	0.017 ± 0.008	0.028 ± 0.022	0.607	0.649	1.000	0.317	
	non-ABS	61	0.023 ± 0.007	0.013 ± 0.004	0.025 ± 0.008	0.520	0.133	1.000	0.077	
DRB	All	79	0.212 ± 0.028	0.141 ± 0.023	0.114 ± 0.025	1.228	0.429	0.202	1.000	
	ABS	15	0.524 ± 0.057	0.594 ± 0.132	0.128 ± 0.080	4.640	0.001	0.001	1.000	
	non-ABS	64	0.144 ± 0.025	0.076 ± 0.015	0.112 ± 0.029	0.678	0.247	1.000	0.129	
Notes.

aa distance average pair-wise amino acid distance;

dN: nonsynonymous, dS: synonymous, Z test p-values for rejecting the null hypothesis of neutrality (dN = dS) for the alternative hypotheses of non-neutrality (dN ≠ dS), positive selection (dN >  dS), and purifying selection (dN < dS).

The results from the selection analyses in PAML revealed different levels of selection for the four loci (Table 4). The variable evolutionary rates across the codon sites (M3) fit our data better than the M0 model and models M2a and M8 had higher log-likelihoods than positive selection (M1a and M7). The M2a and M8 models implied that approximately 2% of sites may be under positive selection at the DQA site (ω = 8.583, ω = 8.425) (Table 4). The posterior means of ω were estimated across the DQA codons under positive selection models and predicted fourteen sites (positions 10, 17, 18, 30, 46, 51, 52, 60, 61, 65, 68, 69, 70, 72) that may be under selection (ω > 1), nine (10, 30, 46, 51, 52, 61, 65, 68, and 72) of which were also putative ABS, based on the HLA equivalents (Fig. 4). However, the discrete model (M3: 3 discrete evolutionary rate classes) had the highest log-likelihood and estimated that only 6.6% of codon sites had ω values greater than one (ω = 10.031) with the remaining 93.4% of sites being assigned ω values close to 0 (Table 4).

Figure 4 The ABS binding residues of DQA (A), DQB (B), DRA (C) and DRB (D) in GZP.

The non-ABS region was circled in white color, the equivalent position of ABS were in red (Non-polar R-amino acids), blue (polar R-amino acids), green (positively charged R-amino acids) and purple (negatively charged R-amino acids), respectively.

However, the posterior means of ω across DQB codon sites estimated by M2a (ω = 6.240) and M8 (ω = 6.373) predicted that only five codons were under significant positive selection (positions 16, 27, 47, 57, 61). These five codons were also known as putative ABS based on the HLA equivalents (Fig. 4). The M3 model at the DQB estimated that approximately 8% of codon sites had ω values greater than one (7.3% with ω = 1.283; 0.6% with ω = 6.935) with the remaining 92% of sites being assigned ω values close to 0 (ω = 0.054) (Table 4).

The M3 model estimated that only 6.6% of codon sites had ω values greater than one (ω = 10.031) with the remaining 93.4% of sites being assigned ω values close to 0 for the DRA (Table 4). Moreover, the posterior means of ω across the DRA codon sites estimated by M2a (ω = 10.286) and M8 (ω = 10.323) predicted that nine codons (positions 12, 14, 15, 16, 18, 19, 49, 64, 68) were under significant positive selection. However, only two sites (19 and 49) were known as putative ABS based on the HLA equivalents (Fig. 4).

The M3 model estimated that 14.6% of codon sites had ω values greater than one (ω1 = 1.351, ω2 = 6.823) at the DRB (Table 4), which was higher than the other MHC class II codons (DQA, DQB and DRA). The posterior means of ω across the DRB codon sites were estimated by the M2a (ω = 5.972) and M8 (ω = 5.961) models and predicted that twelve codons (positions 1, 2, 23, 28, 47, 48, 58, 61, 62, 65, 69, and 77) were under significant positive selection, seven (2, 28, 48, 58, 61, 69, and 77) of which were also putative ABS based on the HLA equivalents (Fig. 4).

Evolutionary analysis

We could not determine the genealogy of DQA, DQB, DRA and DRB due to the presence of loops in the network (Figs. 5A–5D). Some alleles were more likely to be ancestral based on their internal position in the network and a greater frequency of mutational connections. These alleles seemed more likely to be ancestral at the DQA locus, including DQA1, DQA3, Eqca17, and Eqca18. Allele DQA1 appears to be ancestral for most alleles, namely DQA12, DQA13, DQA14, DQA15, DQA9, Eqca10, Eqbu6, Eqca20, Eqas1, Eqas2, Eqbu5, Eqca19, Eqbu4, and Eqbu12. Three haplotypes of Przewalski’s horse (Eqpr3, Eqpr4 and Eqpr2) were separated from DQA3 by two mutational steps and were most closely related to GZP haplotypes. Meanwhile, the DQA1 allele was shared among four species (Eqca15, Eqgr1, Eqbu2 and Eqze1), DQA2 was shared with Eqca14, and DQA3 was shared between two species (Eqca16 and Eqbu20). At the DQB locus, allele DQB1 appears to be ancestral for most alleles, including DQB31, DQB23, DQB32, DQB10, DQB30, DQB33, DQB29, DQB3, DQB24, DQB34, DQB40, Eqas7 and Eqas4. We found that haplotypes DQB5 and DQB13 were shared between Eqca1 and Eqca7, respectively. Allele DRA5 was shared between Eqbu7 and Eqca5, and DRA1 was shared with DRA3 for the DRA locus. Interestingly, allele DRA1 seemed more likely to be ancestral, containing twenty alleles, including DRA21, DRA15, DRA13, DRA8, DRA18, DRA7, DRA14, DRA6, DRA20, DRA19, DRA17, DRA16, DRA2, DRA11, DRA10, DRA12, Eqca2, Eqca6, Eqca7, and Eqca8. Haplotypes Eqhe and DRA9 were separated from DRA1 by as two mutations step as are most closely related GZP haplotypes. Allele DRA5 seemed more likely to be ancestral, Eqbu, Eqze, Eqgr, Egas were separated from DRA5 by one or two mutational step and are most closely related to the GZP haplotypes. Most DRB alleles were dispersed throughout the whole network, and there was a closer genetic relationship between GZP and other horse species. Wild ass haplotypes, Eqas3, Eqas4 and Eqas6, were separated from DRB28 by one mutational step, as are most closely-related GZP haplotypes. Furthermore, the haplotypes DRB2 (Eqpr1) and DRB3 (Eqpr2) were shared by GZP and Przewalski’s horse. The haplotypes DRB1 (Eqca5), DRB2 (Eqca12), DRB4 (Eqca7), DRB5 (Eqca1), DRB15 (Eqca2) and DRB23 (Eqca8) were shared between GZP and the European horse.

Table 4 Estimation of codon evolution models for the ELA class II DQA, DQB, DRA and DRB loci.

Locus	Nested model pairs	p	ln L	Parameter estimates	Site under positive selection	
DQA	M0: one-ratio	2	−1411.41	ω = 1.002	NA	
	M3: discrete	6	−1,286.74	ω0 = 0.000, p0 = 0.585	NA	
				ω1 = 0.283, p1 = 0.361		
				ω2 = 6.020, p2 = 0.054		
	M1a: nearly neutral	3	−1,336.18	ω0 = 0.000, p0 = 0.851	NA	
				ω1 = 1.000, p1 = 0.149		
	M2a: positive selection	5	−1,281.72	ω0 = 0.000, p0 = 0.775	10,17,18,30,46,51,60,61,65,69,70,72	
				ω1 = 1.000 p1 = 0.205		
				ω2 = 8.583, p2 = 0.020		
	M7: beta	3	−1,338.52	p = 0.008, q = 0.054		
	M8: beta& ω	5	−1,281.73	p0 = 0.979, p1 = 0.021	10,17,18,30,46,51,52,60,61,65,68,69,70,72	
				p = 0.005, q = 0.020, ω = 8.425		
DQB	M0: one-ratio	2	−1,682.05	ω = 0.449	NA	
	M3: discrete	6	−1,461.64	ω0 = 0.054, p0 = 0.921	NA	
				ω1 = 1.283, p1 = 0.073		
				ω2 = 6.935, p2 = 0.006		
	M1a: nearly neutral	3	−1,508.29	ω0 = 0.025, p0 = 0.951	NA	
				ω1 = 1.000, p1 = 0.048		
	M2a: positive selection	5	−1,462.34	ω0 = 0.043, p0 = 0.913	16,27,57,61	
				ω1 = 1.000, p1 = 0.081		
				ω2 =6.240, p2 = 0.005		
	M7: beta	3	−1,519.53	p = 0.008, q = 0.054	16,27,47,57,61	
	M8: beta& ω	5	−1,467.39	p0 = 0.994, p1 = 0.005		
				p = 0.072, q = 0.490, ω = 6.373		
DRA	M0: one-ratio	2	−632.76	ω = 0.778	NA	
	M3: discrete	6	−612.55	ω0 = 0.000, p0 = 0.924	NA	
				ω1 = 0.000, p1 = 0.010		
				ω2 = 10.031, p2 = 0.066		
	M1a: nearly neutral	3	−626.11	ω0 = 0.000, p0 = 0.687	NA	
				ω1 = 1.000, p1 = 0.313		
	M2a: positive selection	5	−612.55	ω0 = 0.000, p0 = 0.934	12,14,15,16,18,19,49,64,68	
				ω1 = 1.000 p1 = 0.001		
				ω2 = 10.286, p2 = 0.065		
	M7: beta	3	−627.68	p = 0.013, q = 0.020	12,14,15,16,18,19,49,64,68	
	M8: beta& ω	5	−612.54	p0 = 0.934, p1 = 0.065		
				p = 0.005, q = 6.831, ω= 10.323		
DRB	M0: one-ratio	2	−1,720.32	ω = 0.772	NA	
	M3: discrete	6	−1,534.47	ω0 = 0.074, p0 = 0.854	NA	
				ω1 = 1.351, p1 = 0.131		
				ω2 = 6.823, p2 = 0.015		
	M1a: nearly neutral	3	−1,580.21	ω0 = 0.030, p0 = 0.913	NA	
				ω1 = 1.000, p1 = 0.087		
	M2a: positive selection	5	−1,535.16	ω0 = 0.054, p0 = 0.827	1,2,23,28,48,58,61,62,65,69,77	
				ω1 = 1.000 p1 = 0.158		
				ω2=5.972, p2 = 0.015		
	M7: beta	3	−1,583.97	p = 0.016, q = 0.104	1,2,23,28,47,48,58,61,62,65,69,77	
	M8: beta&ω	5	−1536.75	p0 = 0.984, p1 = 0.015		
				p = 0.087, q = 0.328, ω=5.961		
Notes.

p, number of free parameters in the ω distribution; ln L, log-likelihood; Model parameter estimates include the nonsynonymous to synonymous rate ratio (ω) and proportion of sites (p) under each ω site class. Sites under selection were predicted by the Bayes Empirical Bayes (BEB) approach: sites inferred to be under positive selection with posterior probabilities >99%.

Discussion

Our study revealed the diversity of the four ELA class II gene regions, DQA, DQB, DRA and DRB, and the contribution of many novel alleles identified in GZP. Our data determined within-species variation using the numbers of alleles. 21 DQA alleles, 45 DQB alleles, 22 DRA alleles, and 31 DRB alleles were unequivocally identified from the GZP.

The DRA locus was relatively well-conserved in four GZP loci compared with the other three loci. The alignments of the DQA, DQB, and DRB genes revealed considerable sequence diversity. However, DRA had a lower nucleotide diversity. Our results are consistent with the level of nucleotide diversity at the genus level for Equus ELA genes as reported by Kamath & Getz (2011). DQB had the highest level of polymorphisms with a ratio of polymorphic sites of 46.08%, this was followed by DRB and DQA. DRA had the lowest level of polymorphisms (15.04%), which was consistent with the results of the DRA locus in dogs (Wagner, Burnett & Storb, 1999), cats (Yuhki & O’Brien, 1997), goats (Takada et al., 1998), and pigs (Chardon, Renard & Vaiman, 1999). The genetic diversity of ELA is reportedly important for immune functions involving the resistance and susceptibility to pathogens (Trowsdale & Parham, 2004) with a probable mechanism of gene selection in the evolution process of the pony (Penn & Potts, 1999).

Figure 5 Median-joining network of DQA (A), DQB (B), DRA (C) and DRB (D) sequences in the Equidae family.

The circle size was proportional to haplotype frequency.

We detected the balancing selection events by determining the rate of non-synonymous/ synonymous substitutions (dN/dS ratio) of nucleotides. Our results revealed a high genetic variability at the DQA, DQB, DRA, and DRB loci. The dN/ dS ratio (dN/ dS = 0.560) at the DRA locus was the lowest, which was similar to the low levels of polymorphisms detected by sequence alignment. It has been established that the number of synonymous substitutions is greater than non-synonymous substitutions due to strong functional and structural constraints on the protein (Kamath & Getz, 2011). The number of polymorphisms at the DRA locus may be attributed to the selective pressure for DRA haplotypes that present pathogenic antigens for the host species more efficiently (Albright-Fraser et al., 1996).

We found nine DQA codons, five DQB codons, nine DRA codons, and seven DRB codons under significant positive selection. The majority of these codons were predicted to be the ABS of ELA. The amino acids under site-specific selection were located on the protein surface based on SWISS-MODEL prediction results (Fig. 4) and were found on the inner surface of the MHC cleft with bound peptides in the antigen presentation (Madden et al., 1995). Several reports indicated that the diversity and nonsynonymous mutations at the ABS could improve the hosts ability to recognize pathogens (Hughes & Nei, 1988; Hughes & Nei, 1989). These data suggest that the different rates of non-synonymous and synonymous substitutions in DQA, DQB, DRA and DRB were closely related to the ABS changes in the GZP. In particular, the dN/ dS ratio in the ABS was greater than that in the non-ABS region at the DQA, DQB and DRB loci, which is common in the Argentine Creole horse (Díaz et al., 2001). The dN/ dS ratio of ABS was higher than the other regions, which may be due to balancing selection (Albright-Fraser et al., 1996), and the positive selection results in MHC polymorphisms (Yang et al., 2000).

The haplotype median network of DQA, DQB, DRA and DRB between GZP and other horses (Eqca, E. callabus; Eqpr, E. przewalski; Eqki, E. kiang; Eqgr, E. grevyi; Eqas, E. asinus; Eqbu, E. burchelli; Eqze, E. zebra; Eqhe, E. hemionus) were analyzed. Among these, several wild ass haplotypes were separated from DQA1, DQB1, and DRB28 by one or two mutational steps and are more closely related to GZP haplotypes. The divergence time between the horse and ass has been estimated to be 0.88–2.3 Ma (Krüger et al., 2005). One E. hemionus haplotype (Eqhe2) was separated from DRA by two mutational steps and is most closely related to GZP haplotypes. It suggested that DQA1, DQA3, DQB1, DRA1, DRA5, and DRB28 may be the oldest alleles. The haplotypes DRB2 and DRB3 were shared between GZP and the Przewalski’s horse at the DRB locus. Przewalski’s horse haplotypes Eqpr3, Eqpr4 and Eqpr2 were separated from DQA3 by two mutational steps. Przewalski’s horse was discovered on the Asian steppes in the 1870s and it is the only surviving species of wild horse in the world (Wakefield et al., 2002). It is thought that the Przewalski’s horse and the domesticated horse populations separated about 45,000 years ago and maintained a certain degree of gene-flow (Der Sarkissian et al., 2015). Some haplotypes were shared between the GZP and European horses, including DQA1, DQA3, DQB5, DQB13, DRB1, DRB2, DRB15, DRB23, DRB4, and DRB5. The allele DQA1 appears to be the ancestor for the three alleles, Eqca10, Eqca20, and Eqca19. Allele DRA1 seemed more likely to be ancestral for four alleles, including Eqca2, Eqca6, Eqca7, and Eqca8. The genes of the domesticated Asian horse may have dispersed into European populations because of the gene flow (Bjørnstad, Nilsen & Røed, 2003). Interestingly, the haplotypes DQA1, DQA3, and DRA5 were shared between the GZP and E.burchelli, E.grevyi and zebra. The divergence time between horses and zebras is estimated to be 0.86–2.3 Ma based on microsatellite trees (Krüger et al., 2005). The common ancestor of all extant forms may have existed about 3.9 Ma, and speciation leading to the zebra, ass, and horse may have occurred within the following 0.5 Ma (George & Ryder, 1986). These data and our results indicated that the GZP is an ancient variety of equid. Additional studies on the GZP may advance our knowledge of unique haplotypes and their roles in the adaptation to local environmental pressures such as the unique pathogenic microorganisms in the mountainous and humid districts in Guizhou province, China.

Conclusion

Nucleotide diversity was detected from exon 2 of ELA-DQA, DQB, DRA, and DRB genes in the GZP using direct sequencing technology. Of those four loci, the DRA locus was relatively well-conserved and possessed the lowest diversity. Many codons in the ABS underwent positive selection, including nine DQA codons, five DQB codons, nine DRA codons, and seven DRB codons. The amino acids coded by selected codons were found on the inner surface of the cleft of the ELA complex and were bound to an antigen peptide. The selected sites may be related to the GZP’s ability to defend against foreign pathogens from the surrounding habitat. Many ancient alleles were detected at the DQA, DQB, DRA and DQB gene regions of GZP. Two older haplotypes of DRB (DRB2 and DRB3) were shared by the GZP and Przewalski’s horse. Two older haplotypes of DRA (DRA1 and DRA5) were separated from Eqbu, Eqze, Eqgr, and Egas by one or two mutations steps, and four older haplotypes of GZP (DQA1, DQA3, DQB1, and DRB28) were closer to the wild ass and Przewalski’s horse by only one or two mutational steps. The indigenous breed, GZP, may have retained ancient haplotyes in ELA genes. There may be a large number of unique haplotypes dispersed in GZP resulting from the long process of ELA molecule evolution. The unique genetic characteristics of GZP have been unclear, undervalued, and confused with other ponies. The genetic uniqueness revealed in our study is helpful to understand its genetic conservation of this ancient variety of pony.

Supplemental Information

Supplemental Information 1 The nucleotide of the DQA, DQB, DRA, and DRB locus

Click here for additional data file.

Supplemental Information 2 Nucleotide composition of DQA, DQB, DRA, and DRB in Guizhou pony

Click here for additional data file.

Additional Information and Declarations

Competing Interests

Author Contributions

Animal Ethics

Data Availability

The authors declare there are no competing interests.

Chang Liu conceived and designed the experiments, analyzed the data, prepared figures and/or tables, authored or reviewed drafts of the paper, and approved the final draft.

Hongmei Lei performed the experiments, prepared figures and/or tables, and approved the final draft.

Xueqin Ran and Jiafu Wang conceived and designed the experiments, authored or reviewed drafts of the paper, and approved the final draft.

The following information was supplied relating to ethical approvals (i.e., approving body and any reference numbers):

All animal procedures were approved by the Institutional Animal Care and Use Committee of Guizhou University (Approval number EAE-GZU-2018-P007) and were conducted in accordance with the National Research Council Guide for the Care and Use of Laboratory Animals.

The following information was supplied regarding data availability:

Raw data is available as Supplemental Files.

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
