# Peer review of "Genetic variation and selection in the major histocompatibility complex Class II gene in the Guizhou pony"

_PeerJ, doi:10.7717/peerj.9889_

## Round 0.1 · original submission · Major Revisions

Your paper has now been assessed by 3 expert reviewers. Whilst they all see merit in your paper, they each list a number of concerns that might be addressed.

These include issues with the phrasing of the introduction, coverage of the relevant background literature, use of scientific terminology, analytical approach, and general reporting style.

I am willing to consider a revision of your paper, but note that the changes required are substantial, including reanalysis as suggested by Rev 2.

Reviewer 1 ·

Basic reporting

The study revealed a high level of diversity present in the Guizhou ponies (GZP) through comprehensive analyses conducted on the MHC class II gene, and positively selected codons were mostly found in ABS, which are linked to the defense ability of GZP against foreign pathogens. In addition, both ancient and species-specific alleles were revealed in the gene regions, which shown evolutionary relationships of GZP with other ponies and horses, and genetic novelties gained during the evolutionary course of GZP since its separation with other ponies. These results are significant and meaningful for GZP evolutionary conservation, and values for considering GZP as unique genetic resources. For this reasons I would like to suggest its acceptance for publication on Peer J if they can address my following minor concerns and suggestions.

Experimental design

The data and analysis methods are solid, and the figures and tables are clear. English writing is also acceptable, albeit some small changes should be made.

Validity of the findings

For a long time, the unique genetic characteristics of GZP are unclear, and thus it is undervalued and frequently hybridized with other ponies. This is definitely deleterious to the genetic resource of this indigenous species in China. The genetic uniqueness revealed in this study is helpful to its genetic conservation and to consider it as evolutionary significant subspecies.

Additional comments

1, Line 27 I suggest to delete the “the pony” in “gene in the pony GZP”, GZP means the pony. And the last sentence in the abstract should be polished.

2, Line 75-76, reference(s) should be added

3, Line 91, it should be clarified that how many individuals are represented by the 50 blood samples.

4, Line 102, the related GeneBank sequences are not clearly clarified.

5, Line 152, what is the means of “1000 sequencing data”? And you mentioned “the effective numbers of alleles were 118”, do you mean the rest 66 alleles are not effective? Why ?

6, Line 157 I just can’t get the point of “considerable sequence diversity within the same genus”. Do you mean that DQA/B, DRB harbor a higher level of sequence diversity than that in other species from the (Equus) genus? I did not find the comparative analyses on sequence diversity on DQA/B and DRB among the species within Equus genus.

7, Line 158-159 you mentioned “In contrast, DRA shown much lower diversity of nucleotides, which is consistent with the level of nucleotide diversity at genus of Equus (Kamath et al., 2011)”, according to my understanding, the diversity of nucleotides in DRA is much lower than that in DQA/B and DRB in GZP, which, however, is comparable with the level of nucleotide diversity in DRA from other species within Equus genus.

8, Line 168, “(38.10%) in DQB alleles”, I think you mean DRB here.

9, Line 176, I do not find the “Brown et al. 1993” in the Reference section.

10, Line 222-223, English writing need to be polished, such as “The variable evolutionary rates across codon sites (M3) provided better fit to the data than the M0 model” will be better in expression like “The variable evolutionary rates across codon sites (M3) fit the data better than the M0 model”.

11, Line 251, the subtitle is inappropriate, both selection and network analyses belong to Evolutionary analysis.

12, Line 284, the authors claim that they found elevated levels of diversity at DQA/B, DRA/B in GZP. However, I did not find any comparative analysis on the sequence diversity of DQA/B, and DRB between GZP and other species except that DRA were compared with other horses, dogs, cats, goats and pigs.

Reviewer 2 ·

Basic reporting

English: Low quality of English throughout the entire manuscript; sometimes difficult to understand what the authors want to say (e.g. two last sentences of Abstract). A revision by a native speaker is recommended.

Introduction: This part is not well structured, it does not provide a convincing rationale of the experimental design. To re-write it, reflecting the points below, would be a good idea.
- it contains incomplete and/or incorrect statements - information about the role of MHC genes in immune responses is not fully correct (line 46) and is very general (“initiation of immune response” – line 50); the professional terminology is not always appropriate (“polymorphism of MHC loci diversity”, line 54);
- differences in the function of MHC class I, II and III molecules are not explained; consequently, it is not clear to the reader, why just exon 2 sequences of MHC class II genes show the highest genetic variation (lines 52-53), why they associate with diseases (not only infections – line 55), and why they have been extensively studied in many vertebrate/mammalian species;
- therefore, their relevance for “evolutionary relevant and adaptive processes”, line 61 is not clear;
- the review of literature does include relevant recent information both on the MHC in general and on the MHC of the domestic horse and of other equids; consequently, the experimental design is based on an obsolete concept (see below);

Structure of the manuscript: The paper has standard structure that conforms to PeerJ standards, including acknowledgments/funding information, ethical statements, declaration on competing interests and authors´ contribution. 11 figures and 2 supplementary tables were submitted as separate files. The supplementary tables were attached as Excell files. All figures are labelled and described. Their relevance is related to the limitations of the experimental design (see below). Sequence data are available in the Supplementary table 1. There is no information whether these data have been submitted to a repository, no accession numbers are available in the text.

Experimental design

Originality of the research: The genetic diversity of MHC class II genes and their exon 2 sequences has been studied in many mammalian populations, including horses and other equids. The use of a local pony breed for this purpose represents an original aspect of this study, which is within the scope of the journal.

Research concept: The experimental design and the resulting research question are not based on a current knowledge of the genetic structure of the equine MHC. Important information, especially the data from Viļuma A, Mikko S, Hahn D, Skow L, Andersson G, Bergström TF: Genomic structure of the horse major histocompatibility complex class II region resolved using PacBio long-read sequencing technology. Sci Rep. 2017 Mar 31;7:45518. doi: 10.1038/srep45518, have not been taken into consideration, as well as the last assembly of the domestic horse whole genome sequence (EqCab3.0 – Kalbfleisch et al.: Improved reference genome for the domestic horse increases assembly contiguity and composition. Commun Biol. 2018 Nov 16;1:197. doi: 10.1038/s42003-018-0199-z. eCollection 2018. Erratum in: Commun Biol. 2019 Sep 11;2:342. Commun Biol. 2019 Sep 11;2(1):342.), where the MHC region has been quite well annotated. Other resources, such as Miller et al. (2017) were cited in the context of disease associations, but relevant information related to MHC class II genes was not fully explored. Currently, it is clear that in the domestic horse, the MHC class II region contains multiple DRB, DQA, and DQB genes, including some pseudogenes. Therefore, the major limitation of this manuscript is that it does not distinguish between individual genes. The primers used for amplifying DRB, DQA, and DQB sequences are not locus specific. As only one DRA gene has been identified in horses so far, this problem does not concern this locus. However, for DRB, DQA, and DQB genes, the overall data presented in the manuscript are necessarily biased. As there probably are differences in effects of selections on different loci (e.g. Janova et al. 2009, cited in the paper), this limitation is also important for the analysis of selection. The donkey whole genome sequence (Renaud et al. Improved de novo genomic assembly for the domestic donkey. Sci Adv. 2018 Apr 4;4(4):eaaq0392. doi: 10.1126/sciadv.aaq0392) shows a very similar structure of MHC class II region and multiple class II loci exist probably also in asses and zebras. Taken together, the interpretation of the evolutionary and selection analyses presented here is ambiguous and difficult to interpret.

Methods used:
Important information about the population sampled is missing. It is not clear to what extent the “randomly selected” ponies could be related, and what was the proportion between males and females.
Standard DNA extraction, PCR, and cloning techniques were used. There is no information on how Sanger sequencing was performed (outsourced?). Information on the location of the primers used is not available. Locus-specificity of the primers used has not been assessed.
Evolutionary analyses used standard tools. As for selection analyses, it is not clear how the PAML package was used, especially how decisions about selecting different models were made. Especially, it is not clear whether recombination was considered in this context. As pointed out above, both of these analyses are limited by the experimental design.
It is not clear how, based on HLA equivalents (lines 175-176), the antigen-binding site residues were identified. It seems that only one of the two references “Brown et al.” relates to this problem.
All these issues should be addressed to improve the manuscript. On the other hand, some of the data presented (e.g. Supplementary table 2) are not very informative for the purposes of the article.

Validity of the findings

The effort of the authors has undoubtedly brought information on novel allelic exon 2 sequences of MHC class II DRA, DRB, DQA and DQB existing in Guizhou ponies. As explained above, the interpretation of these data is limited by the experimental design for three out of four sub-regions. The authors do not say how many DRA, DRB, DQA and DQB sequences they retrieved from individual horses. This information would be very helpful for understanding which type of data they actually collected. The sequences obtained have not been deposited in public databases.
For all these reasons, an accurate and correct interpretation of data presented in the manuscript is really difficult, which is also a limitation for the value of Discussion, along with the quality of English.

Additional comments

The manuscript submitted brings new information on genetic variation in exon 2 sequences of MHC class II genes in a local pony breed. Some serious deficiencies across all its sections should be addressed before the data can be published. It probably would not be enough to simply re-write it, but the data obtained could be re-analyzed. If done properly, at least one part of them could be published.
The authors could be encouraged for example by a very recent analysis of a DRB3 locus in cattle (Bohorquez et al Frontiers in Genetics, 2020). The authors also should pay attention to recommendations on comparative MHC nomenclature (Immunogenetics 70: 2018: 625-632).

Reviewer 3 ·

Basic reporting

The paper must be checked carefully for grammatical errors but also for the general English terminology. I have provided some suggestions, but not for all the errors I have found. The general concepts in the introduction/background are quite exhaustive. In some cases, I have provided some suggestions for concepts that could be clarified and expanded as well as for missing references. Article structure appears appropriate, figure relevant to the content and appropriately described and labeled. The manuscript contains the proper analysis and results for the scope of the work and for a publication.

Experimental design

The work has rigorously performed, with scientific and methodological soundness. The aim of the paper was also clearly defined. Methods were sufficiently described with some missing information in the “samples PCR amplification, cloning, and sequencing”.

Validity of the findings

Data were robust and statistically sound. Conclusions were stated appropriately; however, some sentences should be phrased more clearly and improved.

Additional comments

Review on Genetic variation and selection in major histocompatibility complex class II gene in Guizhou pony

This manuscript aims at the characterization of the genetic diversity at MHC II exon 2 of DQA, DRA, DQB, and DRB regions in Guizhou pony (GZP). The relationship between the observed genetic variation at the MHC II of GZP and mechanisms of selection and evolution were also explored. Exon 2 regions of the ELA-DQA, DQB, DRA and DRB genes were amplified and sequenced from a total of 50 samples. Allelic and nucleotide diversity were explored and analysis of nucleotide and amino acid composition performed. Further analyses were performed to characterize the extent of the observed variability. Evidence for selection ware tested with different approaches: dN/dS ratio, Z-test of selection implemented in MEGA7 software. At the four investigated loci, site-specific selection analyses are provided by measuring the relative proportion of nonsynonymous and synonymous substitution along the full amino acid sequence as well as specifically in the antigen binding site (ABS) and non-ABS. Positive selection was further tested using CodeML subroutine implemented in PAML program. Finally, phylogenetic relationships among sequence haplotypes were defined by constructing median-joining haplotype networks. The work was rigorously performed, with scientific and methodological soundness. This study is relevant and shows interesting results. It provides a comprehensive characterization of the genetic diversity at MHC II exon 2 of DQA, DRA, DQB, and DRB regions in Guizhou pony (GZP). It reveals signatures of selection at the immunological relevant antigen binding site (ABS). It gives an interesting overview of the phylogenetic relationships between the Equine leukocyte antigen (ELA) haplotypes. Finally, the manuscript provides a basis for further in-depth investigations on how the observed signatures of selection could be linked to the pathogenic environment to which Guizhou ponies are exposed in the Guizhou province of China.

General comments:
The English language should be improved to ensure that an international audience can clearly understand your text. Some examples include lines 38, 49, 65, 66, 72, 76, 79, 118, 207, 322, 349, 362, 366 – the current phrasing makes comprehension difficult.

L38 I suggest improving the description of lines 38-40. Here, a clearer sentence could highlight the importance of your work.

L46 The major histocompatibility complex (MHC) genes are very well known for their role in adaptive immunity. If you don’t want to be specific then I suggest keeping this sentence general; something like “The major histocompatibility complex (MHC) genes play a major role in vertebrate immune systems”.

L49 Please consider exchanging: “The MHC class II is highly polymorphic cell-surface molecules”: with “The MHC class II are highly polymorphic cell-surface molecules”.

L64 In this sentence I think you can add some more species (and corresponding references) as the mechanisms you are referring have been extensively studied in many other species as well. For instance: correlative and experimental support for the negative frequency-depend selection at MHC genes has been provided in humans (Trachtenberg et al. 2003), reed warbler population (Westerdahl et al. 2004), laboratory mice (Kubinak et al. 2012), stickleback (Eizaguirre et al. 2012; Bolnick and Stutz 2017) and guppies (Phillips et al. 2018). Examples of asymmetric overdominant selection have been showed in a number of natural and laboratory populations (Landry et al. 2001; Richman et al. 2001; Lenz et al. 2009; Schwensow et al. 2010; Lenz et al. 2013), and supported by several computer-based binding prediction studies (Lenz 2011; Lau et al. 2015; Buhler et al. 2016; Pierini and Lenz 2018).
Also, I would suggest some small gramma changes to make the sentence more appropriate “It has been extensively studied that the mechanisms of negative frequency-dependent and over-dominant selection maintain diversity at MHC genes in …”.

L65 I am not familiar with the term “primary of evidences” please consider exchanging with: “primary sources of evidence”.

L72 “In the family Equidae, the horse MHC class II gene, ie. Equine leukocyte antigen (ELA) class II, located on the short arm of chromosome 20q14-q22” I suggest changing with “In the family Equidae, the horse MHC class II gene, i.e. Equine leukocyte antigen (ELA) class II, is located on the short arm of chromosome 20q14-q22”.

L75 Which are the previous studies? Please provide the proper references. Also in this sentence be sure to clarify and distinguish between the peptide binding groove and the antigen binding site (ABS). Exon2 code for a part of the pocked of the MHC molecules, within this sequence some of the amino acid residues are called antigen binding site (ABS), which are the residues rich in genetic variation (where the nonsyn mutations are observed more often than the syn) and, as you well clarify in the results, those which are thought to be the position in contact with antigens peptide.

L76 “The DQA and DRA encoded the α-chain of ELA class II molecule, and extensive polymorphism of DQA and DRA genes have been determined in European equids” I suggest to change with “The DQA and DRA genes encode for the α-chain of ELA class II molecules, and extensive polymorphism of DQA and DRA genes have been determined in European equids”.

L79 Similar to previous please consider to change the sentence with “The DQB and DRB genes encode the β-chain of ELA class II complex, and high level of DRB and DQB polymorphism have been reported for Arabian and European horses”.

L83 Why did you add this sentence about the body height? There is something relevant about this phenotype? Form my naïve point of view would be nice if you can add why this is important.

L101 If possible please provide the reference for the Primer 5 software.

L103 I am not familiar with the way in which you report the PCR details. Could you, for instance, specify the concentration and type of the different reagents in detail (buffer, dNTP, polymerase, and primers)?

L118 I have the feeling that a part of the verb is missing in the sentence.

L189 The first sentence of this paragraph is not very clear “not all the variable amino acids were selected”. I am not familiar with this test but, is the Wu-Kabat index a test for selection? or is a measure for variability that can elucidate positions that are more variable in your comparisons? Introducing the Wu-Kabat index could be very helpful for the readers.

L207 Please consider exchanging with: Our data are in agreement with results observed in the Argentine Creole horse, which exhibited rate…

L216 I think it is better if you report the actual p-value in text and not only in the table, as you did for all the rest of your analysis.

L309 Can you speculate why there is low variability in the DRA locus but you still find nine DRA codons under significant positive selection?

L322 Please consider exchanging: “the evolution relationship” with “the evolutionary relationship”.

L347 I think this sentence is very interesting for the results of your study and you could expand it. For instance, could you explain what “be evolutionarily old enough” means and/or what this implies?

L349 I really enjoy your discussion. I can suggest some small changes “Furthermore, some unique haplotypes of GZP appear gradually in the evolutionary process, and may have a changed under local environmental pressures such as the unique set of pathogenic microorganisms present in the mountainous and humid districts in Guizhou province, China.”

L362 please check separat

L366 I don’t get the meaning of the term crowns here, please check. Also, I would suggest some small changes “Meanwhile, a fairly large number of unique haplotypes dispersed in GZP crowns might have been acquired in the long process of ELA molecule evolution.”

L366 I suggest adding a sentence on to how the study contributes to filling the gap of knowledge in the field.

L559 In Table 4 I cannot find the P capital letter as parameter indicating number of free parameter in the ω distribution. Please double-check this part; as you clam that small p indicates the proportion of site under each (ω) site class.

L631 In Figure 5 I guess the colors here are also indicative of the proportion of time each allele is found in the different species. If yes I think could be good to explain it in the legend.

---

## Round 0.2 · Minor Revisions

Your manuscript has now been reassessed by one of the original reviewers. From my own assessment of the manuscript, and comments from the reviewer, there are a few issues that need addressing, after which I would be happy to recommend your manuscript for publication.

I have attached an annotated version of the manuscript detailing my requests for clarification, along with multiple suggested edits to improve the clarity of language.

Reviewer 3 ·

Basic reporting

no comment

Experimental design

no comment

Validity of the findings

no comment

Additional comments

Thank you very much for all the modifications.
The paper is significantly improved after the changes.
L40 I think the sentence is incomplete.
I would still recommend including a description of the ABS sites in the introduction.

---

## Round 0.3 · accepted · Accept

Thank you for making the extensive stylistic edits requested of your manuscript. I am now happy to recommend it for publication.